# Fungi Inhabiting the Wheat Endosphere

**DOI:** 10.3390/pathogens10101288

**Published:** 2021-10-07

**Authors:** Lidia Błaszczyk, Sylwia Salamon, Katarzyna Mikołajczak

**Affiliations:** Department of Plant Microbiomics, Institute of Plant Genetics, Polish Academy of Sciences, 34 Strzeszyńska Street, 60-479 Poznań, Poland; ssal@igr.poznan.pl (S.S.); kmiko@igr.poznan.pl (K.M.)

**Keywords:** *Triticaceae*, endophytes, mycobiome, wheat–fungal endophyte interaction, epigenetic regulation, endophytic fungi-based bio-substances

## Abstract

Wheat production is influenced by changing environmental conditions, including climatic conditions, which results in the changing composition of microorganisms interacting with this cereal. The group of these microorganisms includes not only endophytic fungi associated with the wheat endosphere, both pathogenic and symbiotic, but also those with yet unrecognized functions and consequences for wheat. This paper reviews the literature in the context of the general characteristics of endophytic fungi inhabiting the internal tissues of wheat. In addition, the importance of epigenetic regulation in wheat–fungus interactions is recognized and the current state of knowledge is demonstrated. The possibilities of using symbiotic endophytic fungi in modern agronomy and wheat cultivation are also proposed. The fact that the current understanding of fungal endophytes in wheat is based on a rather small set of experimental conditions, including wheat genotypes, plant organs, plant tissues, plant development stage, or environmental conditions, is recognized. In addition, most of the research to date has been based on culture-dependent methods that exclude biotrophic and slow-growing species and favor the detection of fast-growing fungi. Additionally, only a few reports of studies on the entire wheat microbiome using high-throughput sequencing techniques exist. Conducting comprehensive research on the mycobiome of the endosphere of wheat, mainly in the context of the possibility of using this knowledge to improve the methods of wheat management, mainly the productivity and health of this cereal, is needed.

## 1. Introduction

Fungi play an essential role in natural ecosystems and in modern agriculture because of their nutritional versatility, miscellaneous lifestyle, and multifarious interactions with plants. Fungi are important decomposers and recyclers of organic materials [1]. They interact with plant roots in the rhizosphere or with aboveground plant components; while living in close association with plants, they are located either outside or within plant tissues [1]. Fungi that periodically or constantly colonize the internal parts of plant tissues without disease manifestation in their host are defined as fungal endophytes [2,3,4]. Fungal endophytes requiring plant tissues to complete their life cycle are classified as “obligate” endophytes. Well-documented examples of obligate endophytes are found among mycorrhizal fungi and members of the fungal genera *Balansia*, *Epichloë*, and *Neotyphodium* from the family Clavicipitaceae (Ascomycota) [5,6]. However, fungal endophytes that mainly thrive outside plant tissues and sporadically enter the plant endosphere are called “opportunistic” endophytes [7,8]. Between these two groups is an intermediate group, which includes the vast majority of endophytic fungi, the so-called “facultative” endophytes [9,10]. 

Fungi that remain endophytic throughout the entire life cycle of the host are categorized as clavicipitaceus endophytes (C-endophytes) and represent class I fungal endophytes [8,11]. Species of clavicipitaceus endophytes, including *Balansia* spp., *Epichloë* spp., and *Claviceps* spp., establish symbioses almost exclusively with grass, rush, and sledge hosts [9,12], in which they may colonize the entire host plant systemically. Members of this class proliferate in the plant shoot meristem, colonizing intercellular spaces of the newly forming shoots, and can be transmitted vertically via seeds [13]. Some *Epichloë* and *Neotyphodium* species may also be transmitted horizontally via leaf fragments falling onto the soil [14].

Fungi that do not remain endophytic throughout the entire life cycle of the host, and furthermore may not be present during the entire life cycle of the host, are categorized as non-clavicipitaceus endophytes (NC-endophytes) and represent the three functional classes: class 2, defined as containing the fungi colonizing above- and below-ground plant tissues, i.e., the rhizosphere, endorhiza, and aerial tissues [15], and being horizontally and/or vertically transmitted [16]; class 3, defined as containing members of the *Dikaryomycota* (*Ascomycota* or *Basidiomycota*) that are mostly confined to the air tissues of various hosts, especially trees, but also other plant taxa [17,18] and are transmitted horizontally [19]; class 4, which comprise dark, septate endophytes, which, similar to mycorrhizal fungi, are restricted to roots, where they reside inter- and/or intracellularly in the cortical cell layers [20]. Detailed information of fungal endophytes classification has been compiled in the review by Rodriguez et al. [11]. 

Endophytic fungal were first isolated from *Lolium temulentum* seeds in 1898 [3]. Currently, no plants have been found without these microorganisms. Endophytes are present in both large trees [21] and lichens [22]. Together with the plants in which they exist, fungal endophytes can occur in various environments: in agricultural and natural, terrestrial [23] and aquatic [24], tropical [25], and high-mountain [25,26]. In addition, endophytes can colonize various plant tissues (Table 1) intercellularly or intracellularly and display various interactions within their hosts. Relationships between the plant and the endophytic fungus can range from beneficial (mutualism or commensalism) to those that are pathogenic to the host plant [8]. Nonetheless, the functions of many of endogenous fungi are still poorly understood. The presence of fungi in the endosphere of plants, regardless of their lifestyle and way of nutrition, is not without significance. It is well known that sterile plants have a reduced vigor [27,28], while introducing endophytic fungi into plant tissue can provide them with many benefits, such as improving growth rates or enhancing defense and immune responses to biotic [29,30] and abiotic stresses [31,32,33]. Therefore, research on the analysis of microbiomes of cultivated plants, including cereals, in order to find beneficial endophytic fungi and formulating supplements for plants based on them is common.

Cereals are critical to global food production and global food security due to their use as an important food for humans and livestock. One of the main cereals is wheat—statistics shows that in the crop year 2018/2019 over 254 million tonnes were produced in Europe, which accounts for 33.9% of the world’s wheat production [34]. However, wheat production in Europe is mainly affected by the occurrence of drought, late spring frosts, and severe winter frosts associated with inadequate snow cover [35]. Recent crop grain breeding programs have made steady improvements in yield quantity and quality, along with biotic and abiotic stress toleration, which have changed the scope and efficiency of wheat breeding strategies [36]. Usually, high productivity of crop grains is accompanied by extensive utilization of agrochemicals for improvement of soil fertility and control of plant diseases, causing drastic effects on the environment and public health. In order to reduce the negative effects of toxic chemicals, there is a continuous global emphasis on sustainable and less chemically dependent organic agriculture. This opens the way for the use of microbial biological control agents, including endophytic fungi. As mentioned above, studying the ecological role of endophytes and understanding the complex interaction between endophytes and host wheat could lead to the identification of symbiotic fungi and, consequently, the design of plant growth biostimulants or a new generation of biological control agents to improve tolerance to the biotic and abiotic stresses of this economically important cereal.

In this paper, we review the literature in the context of the general characterization of fungal endophytes, with a particular focus on studies of wheat endophytes in different varieties and geographic regions. Despite numerous reviews of grass endophytes [5], trees [37], and tropical plants [29], no comprehensive study summarizing the knowledge regarding fungal endophytes and commonly grown cereal plants found in this study exists. We further notice the importance of epigenetic regulation in wheat–fungi interactions and demonstrate the current state of knowledge. Moreover, the possibilities of using symbiotic endophytic fungi in modern agronomy and wheat cultivation are proposed here.

## 2. Isolation of Fungi from the Wheat Endosphere

To study endophytic fungi, culture-based methods [38,39,40,41] and cultivation-independent techniques [42,43,44] are generally used.

Cultivation-dependent techniques are applied to extract fungi growing in plant tissues [38,39,40,41]. Their isolation is mainly based on the fragmentation of plant organs into small fragments, surface sterilization, and then their placement on microbiological agar media [41,45]. The method of plant fragmentation is very popular and is widely used, due to its simplicity and the variety of fungi obtained. On the other hand, limitations of this method also exist, which should be taken into account at the planning stage of the experiments [3,39,43]. Interestingly, Gamboa et al. [45] noticed that endophyte species diversity is negatively correlated with the size of the tissue fragment used for isolation. Thus, more fungal strains can be obtained using smaller pieces of plant tissues. In addition to the size of the plant tissues, factors such as the surface sterilization of plant fragments, the growth media, the incubation conditions for fungal cultures, and the ability of the fungi to sporulate also have impacts on the result of the experiment. A high risk of some fungi, especially those that are less competitive or that grow slowly and can therefore be easily replaced (displaced) by rapidly growing species, being overlooked exists.

In research on wheat endophytes, before surface sterilization, plant samples are usually washed under running water [46,47], and soil residues are removed from the roots, for example by brushing or scraping [48,49]. Ethyl alcohol and sodium hypochlorite are usually used to remove microorganisms from the surface of plant fragments. After sterilization, the tissues are rinsed with distilled water to remove the residues of the reagents used. However, a number of protocols are used in the surface sterilization of plant fragments. Larran et al. [46,47] sterilized wheat leaves, stems, glumes, and seeds via immersion in 96% ethanol for 1 min, sodium hypochlorite (2% available chlorine *v*/*v*) for 3 min, and again in 96% ethanol for 30 s, and finally by rinsing them twice in sterile distilled water. Comby et al. [38] sterilized wheat roots, stems, and leaves by immersing them in 70% ethyl alcohol for 2 min, in 0.5% sodium hypochlorite for 2 min, and in 70% ethyl alcohol for 1 min and finally by briefly rinsing them in sterile distilled water. In the study by Hubbard et al. [32], the wheat seeds were sterilized in 95% ethanol for 10 s, rinsed in sterile distilled water for 10 s, immersed in 5% sodium hypochlorite for 3 min, and then rinsed three times in sterile distilled water. In the work of Bouzouin et al. [41], water-washed wheat root samples were surface sterilized in 75% ethanol for 1 min, and then, after rinsing them in sterile distilled water for 1 min, the roots were immersed in 5% sodium hypochlorite for 3 min and rinsed again in sterile distilled water. Rojas et al. [50] isolated endophytes from wheat kernels. Separated for this purpose, the glumes, lemmas, paleas, and kernels were surface sterilized according to Comby et al. [38]; however, they sterilized flower tissues in 96% ethanol for 1 min, 2% sodium hypochlorite for 3 min, and 96% ethanol for 30 s, and finally rinsed the tissues twice with sterile MilliQ water. In contrast, Cłapa et al. [48] and Salamon et al. [49] developed a protocol for wheat in which plant tissue fragments were rinsed in 70% ethyl alcohol for 30 s and then in 0.5% sodium hypochlorite for 2 min. In order to remove the reagents, the wheat tissues were rinsed several times with distilled water. The sterilization efficiency is usually tested by placing a volume of the last-rinse water on agar plates.

After surface sterilization, the plant material was aseptically cut into smaller pieces 5–10 mm in length [48,49], re-sterilized, and then placed on the agar medium. Many different media can be used, but the most common ones are potato dextrose agar (PDA), malt extract agar (MEA), corn meal agar (CMA), yeast extract peptone dextrose agar, as well as minimal plant tissues or extract media [51,52,53,54]. Typically, the culture medium is supplemented with antibiotics: chloramphenicol, ampicillin, streptomycin sulphate and chlorotetracyclin hydrochloride, and amoxiallin. As in the case of sterilization, many variations of the incubation of plant tissue fragments on a solid medium exist. The temperature range used was 22 C [49]–27 C [47]. Cultures were maintained for a period of 5 days [47] to 28 days [41] or until the mycelium became visible [49], both in the dark [41,50] and in the photoperiod [47,49]. The grown mycelium was then passaged repeatedly onto agar media. However, a laborious and experiential step to obtain pure, monospore cultures via multiple dilutions and the use of micromanipulation techniques was performed. This stage is extremely important for further identification of the fungal isolate as well as for further characterization and use.

## 3. Identification of Endophytic Fungi

Traditionally, morphological features have been used as an approach to identify isolated fungal species, such as observations of mycelium growth in agar media, which enables the assessment of physiological properties such as colony color and growth rate or microscopic observations of spores and spore-producing structures resulting from asexual or sexual reproduction [3]. The techniques light microscopy and scanning electron microscopy make it possible to evaluate the observations of spores in terms of color, shape, and surface type as well as to identify various spore-bearing structures. This approach is still used but requires experienced mycologists. Furthermore, identification based solely on morphological features [55] is not always sufficient, especially when performing identifications at a lower level of classification, e.g., to a species [56,57]. This is the case, for example, due to the high morphological variability of isolates within one species caused by hybridization [58], due to cryptic speciation and evolutionary convergence [59]. In addition to physiology, morphology, or ultrastructure, the tissue biochemistry, ecological features, and chemotaxonomic features of fungi are not always correctly classified using traditional taxonomic methods [59].

The use of molecular techniques such as sequencing methods and the introduction of a DNA barcoding system have overcome the obstacles in traditional identification methods. The DNA barcoding system uses a short and standardized DNA fragment to identify species of microorganisms [60]. Identification is simple when the nucleotide sequence is constant within each species and unique to one species [61]. The most commonly used region for the differentiation of fungi at the genus and species levels is the internal transcribed spacer 1 (ITS1) and 2 (ITS2) flanking the 5.8S rRNA gene. The ITS region is considered very stable, has many copies, and is usually conserved within the species [60,61,62,63,64,65]. The ITS region as a DNA barcode was used in the identification of many fungi important for agriculture, such as *Colletotrichum*, *Fusarium*, *Alternaria*, *Puccinia*, and *Rhizoctonia* [66]. The application of the ITS region as a DNA barcode has many advantages, such as successful amplification among all lineages of fungi using universal primers; suitable fragment lengths; and numerous curated molecular databases in NCBI, UNITE, and EzTaxon. 

However, the ITS barcode has several shortcomings. Various interspecific and intraspecific distances exist between groups of fungi [67], and determining the ITS divergence threshold to distinguish between fungal species is often difficult [68]. In higher taxonomic classifications, the large subunit (LSU, 28S) of rRNA has been shown to have better discriminatory power than ITS as the 28S gene is more variable and is used in classification on the genus to phylum levels [65]. Moreover, using other DNA regions as targets in the methods for identifying fungi is recommended, such as fragments of genes encoding universal proteins: β-tubulin, translational elongation factor 1α, RNA polymerase II, ATP synthase, γ-actin, and calmodulin [56,69,70,71,72,73]. Currently, species identification is built on the basis of a multiloci DNA barcode rather than a single locus. A useful tool for this is multilocus sequence typing (MLST) [74,75,76].

In wheat endophyte research, ITS was the most commonly used DNA barcode in molecular identification, showing some potential in diversity studies or in the search for endophytic strains beneficial to wheat, despite existing limitations in the species discrimination [68,77]. This region of DNA was used only by Larran et al. [47] to identify endophytic fungi isolated from wheat leaves, stalks, chaff, and grains; by Comby et al. [38] for the classification of endoffites from aerial roots and organs, including leaves, stems, anthers, chaff, sediments, and nuclei; and by Bouzouina et al. [41] to determine the species of endophytic fungi isolated from wheat roots. Using high-resolution melting (HRM) techniques and the differences in the melting points of ITS sequences, distinguishing fungi isolated from the inner tissues of wheat plants at the genus level has become possible [48]. However, the multilocus DNA barcode was used by Llorens et al. [78] for correct classification of 2 isolates from the ancestor of wheat, namely *Aegilops sharonensis*; by Salamon et al. [49] for identifying 54 isolates from the root endosphere of common wheat and spelt wheat; and by Rojas et al. [50] to determine the species for 163 fungal isolates from healthy wheat spikes. 

As mentioned at the beginning of this section, the identification of some groups of endophytic fungi can be very difficult, especially those that are closely related to plant tissues and do not grow on standard media or those that, due to their weak substrate competition, may be overlooked in cultures. Culture-independent methods have therefore gained a lot of attention. Among them, meta-barcoding approaches, especially ITS (ITS2) amplicon sequencing, are an important tool that has also been adopted in research on the endophytic fungi of wheat. Recently, this approach has been used by Sun et al. [44], who characterized and compared the communities of fungal endophytes (FEC) from common wheat, wild emmer wheat (*Triticum dicoccoides* Koern.), and sharon goatgrass (*Aegilops sharonensis* Eig). In contrast, Latz et al. [79] analyzed mycobiomes of wheat endospheres to discover the influence of host genotype; abiotic environment (temperature, humidity, and rainfall); and fungi present in the seed material, air, and soil on the formation of endophytic fungal communities in the tissues of wheat plants, along with its growth and development. 

However, ITS sequence-based meta-barcoding has severe limitations in identifying most of the unknown taxa at the species level as many fungi have not been sequenced. Difficulties in correctly identifying taxonomies is also present due to ITS sequence annotations being falsely deposited in GenBank. In addition, for some groups of fungi, the ITS sequences show high inter- and intra-species variability, so the taxonomic assignments with the generally accepted 97% similarity threshold are not consistent for identification at the species level. Taxonomic fungi identification based on high-throughput sequencing can therefore only be justified at the genus or higher levels such as family or order [66,80]. Recently, comparisons of the results of studies on other crops have shown that endophytic fungi discovered by culture-dependent methods differ from those detected by cultivation-independent methods, most puzzlingly, with some isolated strains never having been found by culturing-independent methods [81,82]. This can be explained not only by the variability in the ITS sequence in relation to the sequences of other marker genes or the scarcity of databases, but also by the lack of convergence among taxonomic results, which may be affected by prosaic technical aspects such as the surface of the organ used for analyses; the effectiveness of its surface sterilization; and in the case of high-throughput methods, the effectiveness of tissue maceration and DNA isolation, the amplification reaction; and subsequent stages. Therefore, in research on wheat fungal endophytes, the need to use both approaches to link high-throughput data sets with the results of isolated fungi that are morphologically and phylogenetically identified is worth considering. Among other aspects not discussed here, it at least provides a complete picture of the structure of endophytic fungi in the individual analyzed. For a full insight into the complexity and dynamics of the wheat endosphere mycobiome, the influence of the wheat genotype, the type and age of the organ, the type of tissues, biotic and abiotic environmental factors, and the influence of the remaining microbiome of the studied individual should also be taken into account.

## 4. Assortment and Role of Fungal Endophytes in Wheat

Most fungal endophytes are commensal, have no or an unrecognized effect on the host plant, or show a mutualistic (positive) effect. Such categories of cooperation are known as symbiosis. Interestingly, the type of interaction may be temporary and change under the influence of external factors (e.g., stressful conditions); therefore, endophytes are also latent pathogens and dormant saprobes [8,82,83]. Symbiotic endophytes demonstrate a beneficial impact on their host plant; for example, they can oppose pathogen development [84] by inducing defense mechanisms in their host [85] or by producing antibiotics that inhibit the growth of other microorganisms, including pathogens [86,87]. Moreover, space and resource competition between endophyte and pathogen, or the existence of endophytes acting in a similar way to parasites of plant pathogens within plants were observed [88]. Because several fungi can combine different lifestyles (saprophytic, pathogenic, or symbiotic), their boundaries are often not clear-cut [1]. Many species that are pathogenic for some hosts may be asymptomatic for others [83]. In addition, many fungal endophytes may switch between pathogenic and commensal or mutualistic lifestyles depending on environmental conditions and on the host [37,83]. Based on several investigations, growing evidence suggests that the functions of fungal endophytes and, accordingly, the type of interactions with plants are affected by various abiotic and biotic factors, including environmental conditions, plant genotypes, plant tissue type, the fungal taxon, and strain type, as well as the dynamic network of interactions within the plant microbiome [88]. Nevertheless, the ecological role of endophytic fungi in plants, including wheat, is still poorly understood. 

Research into the distribution and ecological role of fungal endophytes in wheat has been especially intensive in the family Clavicipitaceae, where the asexual genus Neotyphodium and closely related species of the sexual genus *Epichloë* have provided model systems [89]. In contrast with the well-known *Epichloë* and *Neotyphodium* associations with wheat, a lacuna exists in our knowledge of the diversity; the life cycles; and, accordingly, the ecological role of most nonclavicipitaceus endophytic species and the effects of their presence in their wheat host. Nevertheless, the occurrence of endophytic fungi in wheat (*Triticaceae*) has been demonstrated. The characteristics of *Triticaceae* endophytic fungi are summarized in Table 1 and visualized by species and organs of wheat (*Triticum aestivum* L., *Triticum durum*) in Figure 1 and Figure 2. 

The above literature review (Table 1) revealed the possibility of the functioning of various roles of endophytic fungi in relation to wheat, from symbiotic, through saprophytic, to minor or latent pathogenic. Among the species that are considered pathogenic are mainly those of the genera *Fusarium*, *Botritis*, *Cladosporium*, *Septoria*, *Sclerotinia*, *Rhizoctonia*, Pyrenophora, Penicillium, Microdochium, or *Epicoccum*. On the other hand, fungi that show symbiotic interactions with wheat or have a beneficial effect on wheat fitness and yield, or are characterized by antagonistic activities towards its pathogens, are *Trichoderma harzianum*, *Trichoderma hamatum*, *Trichoderma longibrachiatum*, *Rhodotorula rubra*, *Clonostachys rosea*, or *Chaetomium globosum* species. Nevertheless, the role of most fungal endophytes is still poorly understood. It is supposed that species commonly known as pathogens or saprophytes, or those showing mutualistic interactions with plants, may play a completely different role by living in wheat tissue. The function of these species will not be known until the mechanisms of both the unidirectional interactions of these microorganisms with wheat and the complex network of interactions of the entire plant holobiont are known. Therefore, until the appropriate role of the fungus in the phase of its inhabiting the wheat endosphere is known, it should not be classified on the functional level. It is known that fungi have the ability to “switch” their lifestyle and mode of nutrition. Therefore, it is necessary to penetrate the mechanisms of interaction of wheat with endophytic fungi in order to understand the reason for this plant’s “agreement” to inhabit its endosphere. 

The current understanding of fungal endophytes in wheat is built on a rather small set of experimental conditions, including wheat genotypes, plant organs, plant tissues, plant stage development, or environmental conditions [38,53,93,94,99,105,106,107,108,109,110,111,112,113,114]. Furthermore, all of these investigations have been based on culture-dependent methods that exclude biotrophic and slow-growing species and favor the detection of rapidly growing fungi [99,109]. Meanwhile, developments in high-throughput technologies, such as next-generation sequencing (NGS), have opened up new perspectives in fungal endophyte biodiversity research. The pioneers were Nicolaisen et al. [115], who adapted NGS to analyze the mycobiome of 90 wheat grain samples collected from Denmark. Subsequently, Karlsson et al. [106,116] investigated the effect of fungicide use and various crop management practices on the microbiome of the wheat phylosphere in Sweden. Molecular analysis of the ITS region in 220 wheat leaves from 22 fields in Sweden showed an average level of operational taxonomic units (OTU) of 54 and 40 in organic and conventional fields, respectively [116]. Meanwhile, Sapkota et al. [117] conducted a similar analysis on four wheat varieties at two locations in Denmark and identified a total of 212 OTUs. The authors observed that both the geographic location and location of the leaf, genotype, and plant growth stage had an impact on the architecture of the wheat mycobiome. The research of Hertz et al. [118] showed changes in the structure of the mycobiom of wheat ears occurring with their development, and with the time of exposure of plants to biotic and abiotic environmental conditions. Using high-throughput ITS1 sequencing, Yashiro et al. [119] characterized wheat mycobiomes at different stages of cereal processing, comparing domestic environments in rural and urban areas. At a similar time, Vujanovic et al. [108] demonstrated the transgenerational transmission of endophytic seed fungi through three consecutive generations of wheat under control conditions and drought stress, Shiro et al. [120] observed the occurrence of spatial variation in the microbiome of the phyllosphere of commercial wheat crops growing in the same field, and Knorr et al. [121] discovered the effect of fungicide treatments using different dosages, terms, and products on the mycobiom of the wheat phyllosphere. Recently, Latz et al. [79] studied the effects of the host genotype, temperature, humidity and rainfall, and the presence of fungi in the initial seed, air, and soil on the structure of the fungal community inhabiting the wheat endosphere. The studies have shown that the structure of the wheat mycobiome is complex and depends on various elements. A solid evaluation of the factors determining the influence on the wheat microbiome was performed by Kavamura et al. [122]. These factors include: (a) host genotype, growth stage, leaf positions, niche, organs, tissues, hormones; (b) exogenous compounds, namely fungicides, glyphosates, insecticides, phosphine fumigation of stored wheat grains, plastic mulch film residues; (c) fertilization; (d) inoculation of biocontrol agent; (e) land use; (f) management type; g) verhead irrigation; (h) rotation tillage; (i) soil history, type, physicochemical characteristics, and depth; (j) abiotic and biotic stresses; (k) geographical location; (l) growing season. As mentioned, details and literature references on these factors can be found in the review by Kavamura et al. [122].

Due to the recognition of so many factors influencing the structure of the micro- and thus mycobiome of the wheat endosphere, it prompted the search for indigenous species associated with this cereal and forming the so-called “core microbiome” that is constantly associated with a given host genotype. Simonin et al. [123] observed that among 177, 41 fungal taxa were consistently detected in the wheat rhizosphere of African and European soils, constituting a core microbiome. The most frequently detected genera were: *Morteriella*, *Fusarium*, *Exophiala*, and *Chaetomium* [123]. Schlatter et al. [124] described *Nectriaceae*, *Ulocladium*, *Alternaria*, *Mortierella*, and *Microdochium* as core fungal taxa in the rhizosphere of dryland wheat in the Inland Pacific Northwest. Rossmann et al. [125] identified 13 taxa of fungi, namely *Fusarium*, *Fusicolla*, *Purpureocillium*, *Acremonium*, *Bionectria*, *Trichoderma*, *Penicillium*, *Kendrickiella*, *Exophiala*, *Chaetomium*, *Magnaporthiopsis*, and *Staphylotrichum*, corresponding to the core microbiome of wheat cultivated in Brazil. In general, the fungi that typically constitute the core microbiota in wheat are pathogens, mainly of the genus *Fusarium*. Such a generalization, however, would be highly error-prone. In most studies, the core microbiota is defined on the basis of DNA sequence, where, for high-throughput analyzes, the lowest taxonomic unit is at the genus level. Moreover, the core microbiome can be defined in various ways, for example as a component of the microbiome that is constant for the host species over time, or one that determines the functioning of the host species and affects its health and maintenance of homeostasis [126]. Taking into account the functional context on the one hand, and the fact that pathogenic species are listed as components of the core of the microbiome on the other hand, it is worth considering the standardization of its definitions and determination methods. Although no complete understanding of the interactions exists between plants and fungi inhabiting their endosphere, some processes, mainly those involving symbiotic reactions, have been recognized.

## 5. Molecular Interaction between Endophytic Fungi and Wheat

To create and maintain symbiosis, constant communication between the mycobiome and the host plant is required. Sending signals can alter the gene expression and can modulate secreted proteins or metabolites, which have a positive impact on the host [127]. Multi-level interactions are present in the transcriptomes, proteomes, and metabolomes in symbiotic partners. Recent investigation involving Fourier transform infrared (FTIR) spectroscopy and hyperspectral imaging examined the interaction between the endophytic fungal isolate *Penicillium* sp. SMCD 2206 and kernel in durum wheat under drought stress conditions [128]. An altered chemical structure of coleorhizae inoculated with endophyte was observed, which resulted in improved tolerance to drought stress. O–H stretching, acyl lipid chains, proteins, polysaccharide carbohydrates, hemicelluloses, and possibly mannan and glucan may contribute to the chemical differences observed in coleorhizae. Epigenetic mechanisms, including DNA methylation, posttranslational histone modifications, and the activity of small RNAs (sRNAs) and long noncoding RNAs (lncRNAs), alter the chromatin structure and influence the accessibility of genetic information [129]. Such modifications are crucial for blocking the expression of non-genic sequences, such as transposons, repetitive sequences, or pseudogenes in plants [130,131]. Interestingly, epigenetic regulations are induced by environmental signals and can modulate the host–plant interaction with microorganisms and can also control the expression of stress-responsive genes in plants under stress [129,132,133]. Moreover, stress-induced epigenetic changes (epimutations and epialleles) are transient, while others can be stable, maintained, memorized, and transmitted to the next generations [133,134]. Despite the importance of epigenetic control in plant–fungi interactions, the available knowledge concerning wheat plants and endogenous fungi is scarce. However, the ever-increasing availability of high-throughput next-generation sequencing techniques, such as whole-genome bisulfite sequencing (WGBS), small RNA-seq, degradome-seq, etc., as well as recent advances made in wheat genome research [135,136,137] allow the epigenetic control of wheat response to biotic factors to be studied. To draw more attention to this unexplored issue and despite the small number of studies concerning epigenetic control of wheat–fungal endophytes interaction, we performed an extended literature review on the current state of knowledge concerning epigenetic regulation in non-model plant–fungi communication, described below.

### 5.1. Epigenetic Control of Wheat–Fungi Interaction

#### 5.1.1. DNA Methylation

Growing evidence indicates that DNA methylation influences the expression of genes participating in plant response to abiotic and biotic factors. Recent studies suggest that the establishment of endosymbiotic relations is controlled by DNA methylation [132,133]. Beneficial arbuscular mycorrhizal fungus *Funneliformis mosseae* induced changes in the DNA methylation profile in *Geranium robertianum* [134], and conversely, DNA adenine methylation was altered in symbiotic *Mesorhizobium loti* as a result of beneficial relations with their host plants [135]. In wheat, knowledge regarding biotic factor-induced changes in the DNA methylation profile is mainly limited to fungal pathogens. Saripalli et al. [138] observed alterations in the cytosine methylation profiles of susceptible and resistant transgenic wheat lines 96 h after inoculation with biotrophic fungi *Puccinia triticina*, which is the causative agent of leaf rust in wheat. The wheat diploid progenitor *Aegilops tauschii* was used to evaluate the DNA methylation profile during infection with biotrophic fungi, namely *Blumeria graminis* f. sp. *tritici* (Bgt), causing powdery mildew [139]. The authors identified the cytosine methylated in the CHH context as the main loci regulated during the studied interaction, while an expression analysis carried out for certain genes confirmed these findings. However, knowledge concerning the alteration in cytosine methylation caused by endophytic plant symbionts is unexplored. Using methyl-sensitive amplified polymorphism (MSAP), Hubbard et al. (2014) [32] described different DNA methylation patterns in inoculated and uninoculated wheat seedlings in the analyzed conditions. The authors assessed a fungal endophyte referred to as SMCD 2206, which was isolated from surface-sterilized roots of *Triticum turgidum* L., and its role in improving drought and heat tolerance in wheat seeds [31]. Four groups of plants were analyzed: non-stress seedlings without and with SMCD 2206 inoculation, as well as inoculated and non-inoculated plants under drought stress. The DNA methylation pattern in inoculated seedlings under drought was similar to the profile demonstrated by non-stressed samples. Possibly, by changing DNA methylation status in wheat, endosymbiont SMCD2206 differentially expressed crucial genes. This research suggests that endophytes can change the DNA methylation of wheat plants and that the observed changes enhanced wheat resistance against abiotic stresses. Nevertheless, additional studies are needed to confirm these findings as well as to identify the differentially methylated genes that participated in the studied interaction. The question of how much wheat methylome differs in relation to fungi demonstrating different lifestyles or from diverse species still persists. Whether the induced changes are maintained in the next wheat generations and what the role of DNA methylation is in establishing endosymbiotic interactions also remain to be answered. This gap in our knowledge has to be filled in the future. 

#### 5.1.2. Small RNAs 

Endogenous small RNAs (sRNAs) are essential components of the regulatory network of genes participating in host–microorganism interactions. These 20–25 nucleotide-long non-coding RNA molecules repress target gene expression at the transcriptional level via cleavage of the target transcript or at the posttranscriptional level via inhibition of translation [140]. Two types of sRNA molecules can be distinguished: microRNA (miRNA) and short interfering RNA (siRNA). miRNAs are single RNA molecules with stem loop secondary structures, which are encoded by MIR genes located in plant genomes. siRNAs are double-stranded RNA encoded by transposons, viruses, or heterochromatin [141]. In wheat, so far, the involvement of miRNAs during pathogenic fungi infections has been demonstrated [142,143,144,145]. Although the biogenesis pathways of plant miRNAs have been examined in detail and their contribution to the communication between host plants and pathogens has been exposed, knowledge of miRNA participation, regulation, and function in symbiotic plant–fungi interactions is still scarce and limited to the *Medicago truncatula* [146,147], *Solanum lycopersicum* [148], and *Oryza sativa* [149] plant species and to arbuscular mycorrhizal fungi (AMF). However, our very recent study suggests the role of miRNAs in establishing and/or maintaining the wheat–endogenous beneficial fungi interaction (data unpublished). Diverse expression patterns in the roots and leaves of three wheat miRNAs, viz., miR398, miR167, and 159, between *Trichoderma* inoculated (beneficial interaction) or *F. culmorum* inoculated (deleterious interaction) plants and control wheat as well as between *Trichoderma* inoculated and *F. culmorum* inoculated plants were noted. Interestingly, recent studies have discovered that miRNA molecules are transported between plants and microbes and triggered gene silencing as trans-regulators in interacting organisms [150]. The transport of host miRNAs into interacting fungal pathogens has also been observed in wheat–*F*. *graminearum* interactions, where wheat miR1023 suppressed the invasion of *F. graminearum* by targeting and silencing FGSG_03101, which codes an alpha/beta hydrolase gene in *F. graminearum* [150].

#### 5.1.3. Long Non-Coding RNA (lncRNA) 

Transcriptional regulation during wheat–fungi interaction can also be mediated by lncRNAs [151,152,153,154]. This group of non-coding RNAs exceed 200 nt in length and do not contain the significant open reading frame (ORF). Studies on maize indicated that lncRNAs participate in plant–beneficial fungal interactions. Sixty-three differentially expressed lncRNAs were identified in maize under beneficial interaction with the arbuscular mycorrhizal fungi *Rhizophagus irregularis* [154]. In wheat plants, the 254 and 52 lincRNAs (long intergenic ncRNA) responded to pathogenic *B. graminis* f. sp. *tritici* and *P. striiformis* f. sp. *tritici* infections, respectively [151]. The aforementioned report implies that not only pathogenic but also beneficial interaction with fungal symbionts may be controlled by lncRNAs, but more detailed studies are required. 

The studies concerning wheat–fungi interaction should enter an epigenetic era to understand the role of epigenetic regulation in establishing and maintaining the beneficial, endosymbiotic interactions in non-model wheat. Considering the agronomical importance of wheat as well as the lack of detailed knowledge, explorations of the issue presented are needed.

## 6. Application of Endophytic Fungi in Modern Agronomy 

Agriculture today faces the challenge of ensuring food security for the world population, which is estimated to grow from the current level of around 7 billion to 9 billion by 2050 [155]. However, contrary to all opinions, the use of ever greater doses of artificial fertilizers does not increase the yield; on the contrary, this causes a gradual reduction in soil fertility, reduces the quality of cultivated products, and increases environmental pollution. All these aspects have prompted scientists to look for not only an environmentally friendly alternative but also one that would meet the constantly growing demand for agricultural productivity. The interactions of endophytic fungi with crops is of benefit in this regard. These fungi support plant growth and increase tolerance to biotic and abiotic stresses. In sustainable agriculture, endophytes can be used primarily as protection for the host plant against pathogens or pests. They also increase the plant’s resistance to biotic and abiotic stress and affect plant growth and development [29,30,31,32,33,156]. Importantly, they can also support host plants using the metabolites excreted to accelerate the process of nutrient uptake from the environment [157].

The available literature has shown that, among the endophytes identified in wheat, several of them exhibit symbiotic cooperation with this plant. Previous studies by Dingle and McGee [85] on endophytic fungi showed that the *Chaetomium* sp. strain, which was obtained from healthy wheat leaves, contributed to the reduction in the number and development of rust pustules *P. recondite* f. sp. *tritici*. The endophytic strain of *T. hamatum* has been identified as a potential biocontrol agent against *Pyrenophora tritici-repentis* (Died.) Drechsler, the causal agent of tan spots of wheat [156]. Comby et al. [53] found endophytic fungi in wheat that could be used as a biological control agent against *F. graminearum*, the cause of Fusarium head blight (FHB). The identified strains belonged to the following species: *S. kiliense*, *A. proteae*, *C. rosea*, and *M. bolleyi*. Similarly, *S. strictum*, *A. floculossa*, and *P. olsonii* were documented as potential biocontrol agents of Fusarium head blight (FHB) caused by *F. graminearum* in wheat [50]. Disease severity and pathogen biomass inside the analyzed wheat spikes were reduced (70–80%) when the endophytic strains were inoculated at least two days before contact with the pathogen. Interestingly, the endophytic strains used did not present an antagonistic effect on *F. graminearum* during the in vitro dual culture experiment [50]. Furthermore, endophytic *P. olsonii* and *A. alternatum* were identified as biocontrol agents against *Zymoseptoria tritici* causing Septoria tritici blotch (STB) in wheat [79]. Additional inoculation with wheat endophytic fungi also alleviates a wheat plant’s tolerance to salt stress [41]. Under the conditions of moderate salinity, *C. coarctatum* and *A. chlamydospora* intensified the growth of wheat, while under conditions of strong salinity, only *A. chlamydospora* showed this effect. Moreover, *A. chlamydospora* and *F. equiseti* demonstrated the ability to enhance root growth under salt stress [41]. Three endophytes isolated from the roots of *T. turgidum* (referred to as 2206, 2210, and 2215 from the Saskatchewan Microbial Collection Database—SMCD) demonstrated improved tolerance for heat and drought in both parental and second generation durum wheat seeds [31,32]. The authors termed this type of cooperation mycovitality due to the protective fungal effect on seeds, maintaining their vitality and causing successful germination. 

Worth noting is that wheat endophytes are also the substantial source of beneficial metabolite. Pipecolisporin was recently identified in *Nigrospora oryzae* cultures, isolated from *Triticum* sp. roots. This novel compound presents antimalarial and antitrypanosomal activities by exhibiting activity in the molecular range against tropical parasites: *Plasmodium falciparum* and *Trypanosoma cruzi*, respectively [157]. 

The ability to infect cultivated wheat with fungal endophytes that originated in other plant species has also been demonstrated, and various positive effects have been reported. For example, endophytes reduced wheat susceptibility to insects and pathogens [158,159,160], improved heat and drought tolerance [31,32], and promoted plant growth [161]. Serfling et al. [162] have documented the ability of the endophytic species *Piriformospora indica* to reduce common leaf, root, and stem disease symptoms in wheat caused by pathogens such as *Pseudocercosporella herpotrichoides*, *B. graminis* f. sp. *tritici*, and *F. culmorum*. Meanwhile, Malik et al. [163] showed that inoculation with endophytic fungus *Trametes hirsuta*, isolated from the *Chenopodium album* L. plant, may improve the survival of wheat plants in metal-contaminated soils and may additionally assist in the phytoextraction of heavy metals (Pb). Similar properties in relation to wheat plant were revealed for *P. ruqueforti* isolated from the endosphere of *Solanum surattense* [164]. Studies have shown that treating wheat plants grown in soils contaminated with heavy metals Ni, Cd, Cu, Zn, and Pb with *P. ruqueforti* increases their tolerance to stress and nutrient uptake.

The literature review presented above indicates that endophytic fungi isolated from wheat or other plants have much potential to be used in biological control or as plant growth stimulants. However, in order for these microorganisms to be used as bio-pesticides, bio-fungicides, or growth bio-stimulants, they must meet several requirements, such as not being harmful to plants, humans, and animals; effectiveness in controlling their target; the ability to survive in various conditions; and compatibility with the other biologically active substances used in the cultivation of wheat. In addition, their large-scale production should be economically viable [165]. Taking into account the above restrictions as well as the entire commercialization process, which includes the isolation of endophytic fungi; an evaluation of the bioagent’s effectiveness in in vitro, greenhouse, and field conditions; formulation and mass production development; delivery; compatibility; registration; and release [166,167], with large-scale wheat production, bringing such bio-products into the market is a very demanding endeavor. The available literature shows that few such products for wheat have been commercialized so far. Only the following products are documented: AQ10 (Ecogen, Inc, USA) based on *Ampelomyces quisqualis* [167]; Sporodex (Ecogen, Inc, USA) based on *Pseudozyma flocculosa* [167] for protection against mildew powdery; Biomal (Canada) based on *Colletotrichum gloeosporioides* f. sp. *malvae* [168] antagonistic to *Malva pusilla* (round-leaved mallow); Trichodex (Bio works, USA) based on *Trichoderma harzianum* T-39 [169]; Canna based on *Trichoderma afroharzianum* [170]; Trichosan (America) based on the CBS 134709 strain [168] antagonistic to *Botrytis* spp; and Promot WP (USA Canna International BV, NL-Breda, Vitalin Pflanzengesundheit GmbH, D-Ober-Ramstadt JH Biotech Inc., Ventura, CA, USA) based on *Trichoderma simmonsii*, CBS 134706 strain [168], and *Trichoderma guizhouense*, CBS 134707 strain [168], antagonistic to *Fusarium* sp., *Phytophthora infestans*, and *Botrytis* spp.

## 7. New Perspectives and Research Needs 

Significant and continuous technological advances have contributed to the implementation of high-throughput methods over the last ten years for studying the microbiome of various crop species, including wheat. These technological solutions, more precisely NGS, were first used in wheat seed mycobiome research by Nicolaisen et al. [115]. Progress in understanding the complexity of the structure, dynamics, or determinants of changes in the communities of various groups of wheat-associated microorganisms was possible thanks to further research by Karlsson et al. [106,116], Hertz et al. [118], Granzow et al. [112], Gdanetz and Trail [77], Yashiro et al. [119], and Knorr et al. [121]. However, to our knowledge, the first studies of the wheat endosphere mycobiome using high-throughput techniques were published in 2016 by Ofek-Lalzar et al. [39]. The next ones were the work of Vujanovic et al. [108] and Latz et al. [79]. Despite these efforts, knowledge concerning the wheat endosphere mycobiome is still insufficient. Moreover, the synchronization of data obtained over the years and their co-interpretation are problematic, mainly due to the lack of consistent research standards on the wheat microbiome that would allow for a comparison of data from different laboratories, or the revision and integration of data generated from previous methods. With the purpose of mycobiomic research in mind, standardization should include developing the experimental design, adjusting the methodology and strategy for data analysis, interpretation, and integration. When determining the scale, frequency, and time span of sampling, the multidimensional plasticity of the mycobiome should also be taken into account so that both core and transient endophytic fungi in wheat can be identified. Because wheat is a crop, research into its microbiome usually aims to apply knowledge not only regarding the structure but also the function of fungi associated with its tissues. Therefore, comprehensive studies are recommended, including both high-throughput analyses and those based on classical methods enabling the isolation and direct characterization of endophytes. Recently, Kavamura et al. [122] suggested a multi-omic approach for the effective use of the wheat microbiome in efforts to increase the sustainable production of this grain. They proposed metagenomics as describing the structure and diversity of the microbiome, metatrancriptomics for the evaluation of active microorganisms or their genes, and culturomics and phenomics to isolate microorganism of interest and to detect their functional and metabolic activity. However, to prove the functional ability of the selected isolates, Kavamura et al. [122] advised the use of single-cell genomics to target genes of interest. To verify the effect of isolated microorganisms on wheat, they suggest that metaproteomic or metabolomic analyses should be performed on the plant level. Such a comprehensive approach could be adapted to the analysis of the structure and functionality of only the mycobiome of the wheat endosphere, taking into account its multidimensional plasticity. Here, however, a serious obstacle to obtaining complete knowledge is the inability of some endophytic fungi to live outside plant tissues. Therefore, a major challenge for the future is to develop a methodology to capture and maintain such microorganisms outside the plant system. 

In order to obtain a complete understanding of the wheat mycobiom, all the factors that affect its formation and functioning should be taken into account. Kavamura et al. [122] declared that four types of factors determine the microbiome of wheat: antropogenic, edaphic, environmental, and host. When studying the fungi associated with the wheat endosphere, both these factors and the multidirectional interactions of plant–microorganism–microorganism, in line with the latest concept of meta-organisms or the so-called holobiont theory, are worth considering [171]. Currently, endophytic fungi, due to their “type of interaction” with plants or the remaining dead organic matter, are referred to as pathogens, symbiotes, saprophytes, or those whose function has not yet been understood. Until sufficient knowledge regarding the real interactions of these fungi with plants in the endosphere, including the molecular basis of these interactions and their effects on the plant, is presented, this classification should be abandoned. Moreover, the result of the interaction is determined by the entire holobiont of the plant and species, which, apart from this holobiont, exhibits pathogenic features and may perform completely different functions in its endosphere.

Ultimately, research on mycobiomes aims to improve the functioning of the wheat holobiont, stimulating plant germination and growth, providing nutrients, increasing resistance to biotic and abiotic stress factors, and increasing productivity, i.e., yielding. Precision farming, which aims to use a new generation of targeted inoculants based on microorganisms or their metabolites, is a new perspective. Inoculants based on one microorganism or consortia of different microorganisms, closely matched to the host organism or the growing conditions, are considered. The development of such products requires a large amount of research; recognition of their effects on the plant and environment; stability in the target environment; and in the case of microbial consortia, no antagonistic behavior. Such an approach is necessary for the microbiome to be successfully and fully implemented in agriculture and precision farming.

Another perspective based on symbiotic communication between endophytic fungi and wheat is provided by RNAi technologies. This technology uses an internal RNA interference mechanism (RNAi) that occurs in nearly all eukaryotes in which target mRNAs are degraded or functionally suppressed [172]. Two dsRNA delivery strategies are suggested to protect wheat from pathogenic fungal diseases: the transgene-based host-induced gene silencing (HIGS) strategy, which is based on the expression of hairpin RNA or small RNA directed to silence genes in pathogens and pests in a plant, and spray-induced gene silencing (SIGS), which use RNAi-based products such as dsRNA or sRNA that are derived from microorganisms and, accordingly, capable of controlling pests and pathogens present in the phylosphere [170,172]. The latter strategy is considered environmentally friendly and was the first applied by Koch et al. [173], who used foliar application of dsRNA targeting the cytochrome P450 (CYP3) gene in *F. graminearum* and observed reductions in pathogen growth in directly sprayed leaves as well as in distal untreated leaves of barley plants. Unfortunately, in wheat, research on the use of the SIGS strategy is still in its infancy, especially in terms of interactions with symbiotic endophytic fungi.

## 8. Conclusions

Due to the use of wheat as food for humans and livestock, its importance for global food production and global food security and the risks caused by adverse environmental conditions, changing climate, emerging pathogens, and diseases have been extensively discussed. Modern agriculture and the cultivation of wheat therefore face many challenges in order to avoid these threats. One of the ways to reduce the effects of abiotic stresses and to reduce the occurrence of pathogens and related diseases is to use symbiotic endophytic fungi inhabiting the internal tissues of wheat. Numerous studies have been carried out on the use of these microorganisms in biological control or as plant growth biostimulants, although, in the case of wheat, the scope of this work is still insufficient and has not resulted in beneficial solutions in disease management and integrated plant protection. These and the abovementioned studies could contribute to the provision of new tools that can be used for modern agriculture and the management of large-scale wheat cultivation in the world.

## Figures and Tables

**Figure 1 pathogens-10-01288-f001:**
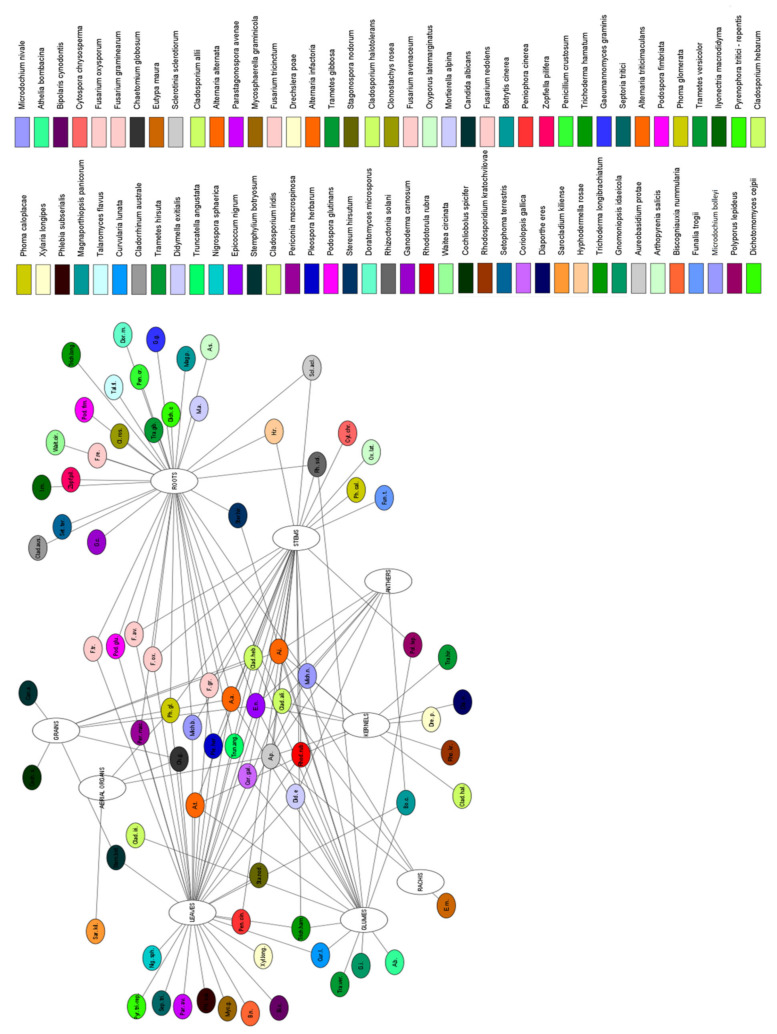
Network of identified endophytic fungal species in the organs of common wheat.

**Figure 2 pathogens-10-01288-f002:**
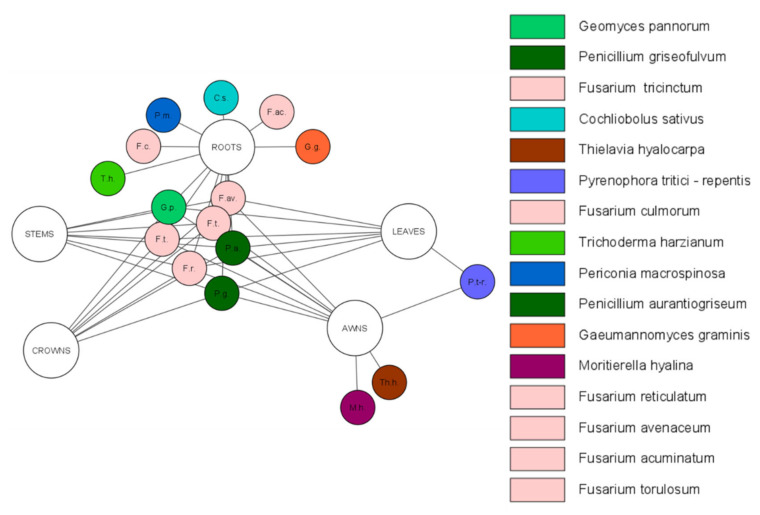
Network of identified endophytic fungal species in the organs of *Triticum durum*.

**Table 1 pathogens-10-01288-t001:** Characteristic of fungal endophytes identified in wheat (Triticaceae).

Species ^1^	Tissue Type	Role ^2^	Localization	Wheat	References
*Alternaria alternata*	roots, stems, leaves	saprophyte/ pathogen	South Africa	*Triticum aestivum*	[46,47,90,91]
	leaves		Argentina		
	leaves, stems, glumes, grains				
*Alternaria infectoria*	rachis, leaves, glumes, anthers, stems, grains	pathogen	France		[38]
	leaves, glumes, grains		Argentina		[53]
*Alternaria triticimaculans*	leaves, glumes, stems, grains	pathogen	France		[38]
*Arthopyrenia salicis*	roots	unrecognized	Poland		[49]
*Athelia bombacina*	glumes	unrecognized	France		[38]
*Aureobasidium proteae*	aerial organs	unrecognized			[92]
	rachis, anthers, stems, grains				[38]
*Bipolaris cynodontis*	leaves	unrecognized	Argentina		[47]
*Bipolaris sorokiniana*	roots, stems, leaves, crowns	pathogen	Canada	*Triticum durum*	[47,53,93,94]
	leaves		Argentina	*Triticum aestivum*	
	stems, grains				
*Biscogniauxia nummularia*	leaves	unrecognized	France		[38]
*Botrytis cinerea*	leaves, glumes, anthers	pathogen			[38]
*Candida albicans*	grains	unrecognized	Argentina		[53]
*Chaetomium globosum*	leaves	unrecognized mycoparasites	France		[38,47,53,92,95,96]
	leaves		Argentina		
	aerial organs		France		
	leaves, grains		Argentina		
*Cladorrhinum australe*	roots	unrecognized	Poland		[49]
*Cladosporium allii*	grains, rachis, roots, leaves, anthers	unrecognized	France		[38]
*Cladosporium cladosporoides*	roots, stems, leaves, awns, crowns	unrecognized mycoparasite	Canada	*Triticum durum*	[93]
*Cladosporium halotolerans*	grains	unrecognized	France	*Triticum aestivum*	[38,92]
	grains				
*Cladosporium herbarum*	leaves	saprophyte/ pathogen	Argentina		[47]
	leaves, stems glumes, grains				[53]
*Cladosporium iridis*	glumes	unrecognized	France		[38]
*Cladosporium minourae*	roots, stems, leaves, awns, crowns	unrecognized	Canada	*Triticum durum*	[93]
*Clonostachys rosea*	roots	unrecognized mycoparasite	France	*Triticum aestivum*	[38,92,97]
*Cochliobolus sativus* (*Bipolaris sorokiniana*)		pathogen	Canada	*Triticum durum*	[93]
*Cochliobolus spicifier* (*Curvularia spicifera*)	grains	pathogen	Argentina	*Triticum aestivum*	[53,90]
*Coriolopsis gallica*	glumes, stems	unrecognized	France		[38]
*Curvularia lunata*	leaves, glumes	pathogen	Argentina		[53]
*Cytospora chrysosperma*	stems	unrecognized	France		[38]
*Diaporthe eres* (*Phomopsis velata*)	grains	unrecognized			
*Dichotomomyces cejpii* (*Aspergillus cejpii*)	roots	unrecognized			
*Didymella exitialis* (*Neoascochyta exitialis*)	leaves, glumes, anthers	pathogen			
*Doratomyces microsporus* (*Cephalotrichum microsporum*)	roots	unrecognized			
*Drechslera poae* (*Pyrenophora poae*)	grains	pathogen			
*Epicoccum nigrum*	roots, stems, leaves	saprophyte/ pathogen	South Africa		[38,46,47,53]
	leaves, anthers, grains	pathogen	France		
	leaves	saprophyte/ pathogen	Argentina Argentina		
	leaves, stems, glumes, grains				
*Eutypa maura*	rachis	unrecognized	France		[38]
*Funalia trogii* (*Trametes trogii*)	stems	unrecognized			
*Fusarium tricinctum*	roots, stems, awns, crowns	pathogen	Canada	*Triticum durum*	[93]
*Fusarium acuminatum*	roots	pathogen			
*Fusarium avenaceum*	roots, stems, leaves, awns, crowns	pathogen			[46,49,93]
	roots, stems, leaves	pathogen	South Africa	*Triticum aestivum*	
	roots	pathogen	Poland	*Triticum aestivum*	
*Fusarium culmorum*		pathogen	Canada	*Triticum durum*	[94]
*Fusarium graminearum*	stems	pathogen	France	*Triticum aestivum*	[38,49,53]
	leaves, stems		Argentina		
	roots		Poland	*Triticum aestivum* spp. *spelta*	
*Fusarium oxysporum*	leaves, stems	pathogen	Argentina	*Triticum aestivum*	[49,53,98]
	roots		Poland		
*Fusarium redolens*		pathogen	France		[38,49]
			Poland		
*Fusarium reticulatum*	roots, stems, leaves, awns, crowns	pathogen	Canada	*Triticum durum*	[93]
*Fusarium torulosum*		pathogen			
*Fusarium tricinctum*	leaves	pathogen	France	*Triticum aestivum*	[38,49]
	roots		Poland		
*Gaeumannomyces graminis*		pathogen	Canada	*Triticum durum*	[93]
			France	*Triticum aestivum*	[38]
*Ganoderma carnosum*		unrecognized			
*Geomyces pannorum* (*Pseudogymnoascus pannorum*)	roots, stems, leaves, awns, crowns	unrecognized	Canada	*Triticum durum*	[93]
*Gnomoniopsis idaeicola*	glumes	unrecognized	France	*Triticum aestivum*	[38]
*Hyphodermella rosae*	roots, stems	unrecognized			
*Ilyonectria macrodidyma* (*Dactylonectria macrodidyma*)	roots	unrecognized			
*Magnaporthiopsis panicorum*	roots	unrecognized	Poland	*Triticum aestivum* spp. *spelta*	[49]
*Microdochium bolleyi*	roots, stems, leaves	unrecognized pathogen	South Africa	*Triticum aestivum*	[38,46,49,92,99]
	roots		France		
			Poland		
*Microdochium nivale*	roots, leaves, glumes, stems, anthers, grains	pathogen mycoparasite	France		[38,100]
*Mortierella hyalina*	awns	unrecognized	Canada	*Triticum durum*	[93]
*Mortierella alpina*	roots	unrecognized	France	*Triticum aestivum*	[38]
*Mycosphaerella graminicola* (*Zymoseptoria tritici*)	leaves	pathogen			
*Nigrospora sphaerica* (*Nigrospora oryzae*)		unrecognized	South Africa		[46]
*Oxyporus latemarginatus*	stems		France		[38,101]
*Parastagonospora avenae*	leaves	unrecognized/ pathogen			
*Penicillium aurantiogriseum*	roots, stems, leaves, awns, crowns		Canada	*Triticum durum*	[93]
*Penicillium crustosum* (*Penicillium solitum*)	roots	unrecognized	Poland	*Triticum aestivum* spp. *vulgare*	[49]
*Penicillium griseofulvum*	roots stems, leaves, awns, crowns		Canada	*Triticum durum*	[93]
*Peniophora cinerea*	leaves, stems		France	*Triticum aestivum*	[38]
*Periconia macrospinosa*	roots		Canada	*Triticum durum*	[38,49,93]
	roots, leaves		France	*Triticum aestivum*	
	roots		Poland		
*Phlebia subserialis*	leaves		France		[38]
*Phoma caloplacae* (*Diederichomyces caloplacae*)	stems				
*Phoma glomerata* (*Didymella glomerata*)	roots, stems, leaves	saprophyte/ pathogen	South Africa		[46,92]
	aerial organs		France		
*Pleospora herbarum* (*Stemphylium vesicarium*)	roots, stems, leaves	saprophyte/ pathogen	South Africa		[46,47,53]
	leaves		Argentina		
*Podospora fimbriata* (*Schizothecium fimbriatum*)	roots	unrecognized	France		[38]
*Podospora glutinans* (*Schizothecium glutinans*)	roots, leaves				
*Polyporus lepideus*	glumes, stems				
*Pyrenophora tritici-repentis*	leaves, awns	pathogen	Canada	*Triticum durum*	[93]
	leaves		France	*Triticum aestivum*	[38]
*Rhizoctonia solani*	stems, roots, glumes				
	roots		Poland		[49]
*Rhodosporidium kratochvilovae* (*Rhodotorula kratochvilovae*)	grains	unrecognized	France		[38]
*Rhodotorula rubra* (*Rhodotorula mucilaginosa*)	leaves	unrecognized/ mycoparasites	Argentina		[47,53,102]
	leaves, stems, glumes				
*Sarocladium kiliense*	stems	unrecognized	France		[38,92]
	aerial organs				
*Sclerotinia sclerotiorum*	roots, stems	pathogen			[38]
*Septoria tritici* (*Zymoseptoria tritici*)	leaves	unrecognized/ pathogen	Argentina		[53,103]
*Setophoma terrestris*	roots	unrecognized	Poland		[49]
*Stagonospora nodorum* (*Parastagonospora nodorum*)	stems, leaves	pathogen	South Africa		[46]
*Stemphylium botryosum*	leaves, grains	unrecognized	Argentina		[53]
*Stereum hirsutum*	roots, glumes	unrecognized	France		[38]
*Talaromyces flavus*	roots	unrecognized			
*Thielavia hyalocarpa* (*Cladorrhinum hyalocarpum*)	awns	unrecognized	Canada	*Triticum durum*	[93]
*Trametes gibbosa*	roots	unrecognized	France	*Triticum aestivum*	[38]
*Trametes hirsuta*	grains	unrecognized			
*Trametes versicolor*	glumes	unrecognized			
*Trichoderma harzianum*	roots	mycoparasites	Canada	*Triticum durum*	[93,104]
*Trichoderma hamatum*	leaves, stems, glumes	unrecognized/ mycoparasite	Argentina	*Triticum aestivum*	[53,104]
*Trichoderma longibrachiatum*	roots	unrecognized/ mycoparasite	Poland		[49,104]
*Truncatella angustata*	roots, stems, leaves	unrecognized	South Africa		[46]
*Xylaria longipes*	leaves	unrecognized	France		[38]
*Waitea circinata*	roots	unrecognized	Poland		[49]
*Zopfiella pilifera*	roots	unrecognized			

^1^—name of the species indicated in the references and valid (in brackets) according to Index Fungorum; ^2^—the role of the species indicated only in relation to wheat or its pathogens (for antagonistic fungi, mycoparasites).

## Data Availability

Data sharing not applicable. No new data were created or analyzed in this study. Data sharing is not applicable to this article.

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
