# Peer review of "Fungi Inhabiting the Wheat Endosphere"

_pathogens, 2021, doi:10.3390/pathogens10101288_

Round 1
Reviewer 1 Report
This is a review paper about fungal endophytes in wheat, where the importance of endophytes and wheat is first described in general, then the different methodologies for the isolation and identification of endophytes are detailed, with special attention to those performed on wheat studies. Afterwards, the role of different endophytic fungi on wheat is analyzed (with the inclusion of a list with the endophytes recorded in wheat), and then, a molecular approach to know the mechanisms involving in the interaction between endophyte-plant is described. Finally, the current applications of endophytes in agriculture (especially focused on wheat) are described. In this part, new perspectives are also included.
I think the authors have successfully reflected the current state of the art about the suggested topic, including most of the relevant and updated bibliography on the subject. In my opinion the main novelty and strengthened part are included in sections 4 and 5. Conversely, the sections 2 and 3 are mostly already known parts, with not much novelty. Although it is also true that such a part is not very long and it is mainly focused on wheat. Also, as indicated in the reviewed version of the manuscript, a new section entitled: 6. New perspectives and research needs, could be interesting to be created. In this new section, the final part of the current section 5 (perspectives) and most of the aspects indicated in the current conclusions section (the part of needs) should be included. The conclusions should be focused on the current knowledge instead.
The manuscript is presented clearly and it is easy to read. Further aspects have been directly included in the reviewed version of the manuscript. Also, although I am not a native English speaker, I feel like the English of the manuscript need to be improved.

Author Response
Thank you for your opinion and friendly comments.
We have made many changes to the manuscript to meet all the expectations and suggestions of the Reviewers and to make the current version attractive, more informative, presenting the latest developments in the field and our critical view.
We hope the manuscript will be acceptable for publication in “Pathogens” in its current version.

Reviewer 2 Report
The presented manuscript by Błaszczyk et al. is reviewing the research on fungi inhabiting the wheat endosphere. Special accent is given on the fungal-plant communities, taking into account the latest work exploring the role of epigenetic modifications in these interactions. Finally, the authors discussed a potential application of endophytic fungi in agriculture, focusing on wheat management. This work is comprehensive, easy-to-follow, and up-to-date with literature. Only minor English editing is required.
Author Response
Thank you for your opinion and friendly comments.
Nevertheless we have made many changes to the manuscript to meet all the expectations and suggestions of the Reviewers and to make the current version attractive, more informative, presenting the latest developments in the field and our critical view. We also improved the English spelling by taking advantage of the professional language service offered by MDPI. The changes made are now visible in Word Review with full customization.
We hope the manuscript will be acceptable for publication in “Pathogens” in its current version.

Reviewer 3 Report
Authors’ effort to write a review on 'Fungi inhabiting the wheat endosphere' is appreciated as such topic has received little attention. This however also means that there are limited research and knowledge on fungal endophytes on non-model plants, especially agricultural crops. Endophytism itself is a difficult research area due to the confounding overlaps of fungal lifestyles – saprophyte, endophyte and pathogen. The fact that fungi can have transitional phases between these lifestyles depending on the conditions of the environment and interactions with the host certainly complicates the study. Despite the challenge, over the years, with the advancement of high-throughput next generation sequencing and molecular technologies, we do know more about these inhabitants compared to decades ago.
The manuscript in its current form is not suitable for publication due to the following reasons:
- Lack of interpretations between numerous studies and authors constantly draw the conclusion of ‘poorly understood’. In a review article, it is important that the authors sum up the current state of the research on this particular topic, and should provide major advances and discoveries, and the significant gap in the research. Authors should perhaps be less apprehensive when making summary but rather provide a more critical view and opinion. For examples, section 1 – what’s the relevance of the various methodologies applied by the different studies and the effect on the recoveries of these endophytes? Review articles are not just compilations, authors are welcome to make critical and judgemental opinions.
- Although this manuscript is supposed to be a review article on wheat, there are inadequate literature survey done on this crop. There are certain sections where authors struggled to provide suitable context due to the lack of relevant literatures. Example, Subheading 3, 4.1
- There are some sections that have irrelevant context to this subject – Line 464 – 479.
- For this type of review article, it is important to address the philosophical question between the differences of endophytic and pathogenic phases especially in wheat because although there are not many research done on endophytes, research on wheat fungal pathogens are plentiful. Due to this, there will be some occurrences of bias in the literature survey with observations and opinions drawn from research on fungal pathogens rather than fungal endophytes. Strictly pathogenic pathogen should not be included in this review – in my opinion. Fungal species that have been discovered from wheat are so diverse but authors need to be clear when interpreting studies from pathogenic species, and of those with latent pathogenic phase, or those that co-exist in a complex. Species such as Fusarium is a good example where you have both endophytes and pathogenic strains. Alternaria sp. is another one.
- Subsection 5 is probably the most relevant to this topic however overall this manuscript could benefit with a focus on the ecological significance as well as the highly diversify mycobiota, and the benefit of fungal endophytes in improving host fitness against biotic and abiotic stress. The text though is voluminous, however it appears to wander away from the scope of the main objective.
Other minor comments:
Structure of some sentences need to be check. For example Line 77 – 80, 252 – 255, 345 – 348, 464 - 466
Line 245 – 247 Sentence was duplicated.
Author Response
Thank you for your feedback, critical comments and very inspiring suggestions. We have made many changes to the manuscript to meet all the expectations and suggestions of the Reviewers and to make the current version attractive, more informative, presenting the latest developments in the field and our critical view.
We hope the manuscript will be acceptable for publication in Pathogens in its current version.

Reviewer 4 Report
The review describes wheat endophytes, their role in plants, some genetic bases of the host-fungis interaction, and prospects of their use in agricultural production. Authors made a good and detailed presentation of the problem and the objects of the study and a good substantiation of the purpose of the review. In general, the review is quite good and can be interesting for readers. English is quite good and understandable, though some language editing is required. I have some minor comments across the text and one concerning Table 1 (see below) that requires some work.
Comments:
Line 53: doubled “categorized”.
Line 83: statistics showS
Line 114: …their sterilization… - probably, surface sterilization?
Line 114-115: “laboratory medium” – I’ve never meet such term in papers. Maybe it would be better to replace this phrase here and in other parts of the manuscript with the more common “agar medium” (or agarized medium, agarized nutrient medium, etc.)?
Line 184: “wheat spelled”? What is this? A misprint?
Table 1: why such species as Fusarium avenacium, F. culmorum, and F. graminearum are considered as latent pathogens?? They cause economically important wheat diseases. For example, Fgra as well as other two species causes Fusarium head blight. Fgra and Fcul also cause Fusarium crown and foot rots, another common diseases of wheat in dry regions.
The same is for Septoria tritici. This is not “unrecognized”, this is a dangerous wheat pathogen causing significant wheat losses
https://www.apsnet.org/edcenter/disandpath/fungalasco/pdlessons/Pages/Septoria.aspx
Stagonospora nodorum is another important wheat pathogen causing Septoria nodorum blotch.
There can be also another real pathogens in the table… I mentioned only those, which are recognized by plant pathologists as real pathogens and are studied by my colleagues who work with the most important wheat diseases. Therefore, I would recommend authors to thoroughly check this table and to correct the role of some fungi if necessary.
Fig. 1: please, check the circles of dark colors: some of them are too dark to read the text within them. The font size is too small to read, but the figure magnification results in a drastic reduction of the image quality; probably, this problem also should be fixed.
Line 348: "...and triggered gene silencing in trans in interacting organisms..." - what is "trans"?
line 367: "...and the significant open reading frame (ORF) is absent" - please, check this phrase, it seems a word is missed.
Line 431-432: ... to reduce common symptoms of leaf, stem base, and root in wheat??? Please, check the sentence.
Author Response

(The authors gave the same response as above.)

Round 2
Reviewer 3 Report
Check format of manuscript - uneven spacing
Author Response
Dear Reviewer 3,
Thank you very much for your attention regarding the manuscript form. We checked the formatting of the entire manuscript, including the table. The changes are listed in the manuscript and visible next to the option - full adjustment.
Lidia Błaszczyk